# Biogenic Gold Nanoparticles Decrease Methylene Blue Photobleaching and Enhance Antimicrobial Photodynamic Therapy

**DOI:** 10.3390/molecules26030623

**Published:** 2021-01-25

**Authors:** Irena Maliszewska, Ewelina Wanarska, Alex C. Thompson, Ifor D. W. Samuel, Katarzyna Matczyszyn

**Affiliations:** 1Department of Organic and Medicinal Chemistry, Wrocław University of Science and Technology, Wybrzeże Wyspiańskiego 27, 50-370 Wrocław, Poland; ewelina.wanarska@pwr.edu.pl; 2Organic Semiconductor Centre, SUPA, School of Physics and Astronomy, University of St Andrews, St Andrews KY16 9AJ, UK; at226@st-andrews.ac.uk (A.C.T.); idws@st-andrews.ac.uk (I.D.W.S.); 3Advanced Materials Engineering and Modelling Group, Wrocław University of Science and Technology, Wybrzeże Wyspiańskiego 27, 50-370 Wrocław, Poland

**Keywords:** antibacterial PDT, antibiotic resistance, methylene blue, biogenic gold nanoparticles, LED

## Abstract

Antibiotic resistance is a growing concern that is driving the exploration of alternative ways of killing bacteria. Here we show that gold nanoparticles synthesized by the mycelium of *Mucor plumbeus* are an effective medium for antimicrobial photodynamic therapy (PDT). These particles are spherical in shape, uniformly distributed without any significant agglomeration, and show a single plasmon band at 522–523 nm. The nanoparticle sizes range from 13 to 25 nm, and possess an average size of 17 ± 4 nm. In PDT, light (from a source consisting of nine LEDs with a peak wavelength of 640 nm and FWMH 20 nm arranged in a 3 × 3 array), a photosensitiser (methylene blue), and oxygen are used to kill undesired cells. We show that the biogenic nanoparticles enhance the effectiveness of the photosensitiser, methylene blue, and so can be used to kill both Gram-positive (*Staphylococcus aureus*) and Gram-negative (*Escherichia coli*) bacteria. The enhanced effectiveness means that we could kill these bacteria with a simple, small LED-based light source. We show that the biogenic gold nanoparticles prevent fast photobleaching, thereby enhancing the photoactivity of the methylene blue (MB) molecules and their bactericidal effect.

## 1. Introduction

The risk of infection in patients from contaminated areas has been discussed in healthcare for many years. Across the world, the cleaning process itself is the subject of widespread attention, concerning methods, equipment, benchmarks, monitoring, and standards for surface cleanliness [1]. In recent years, many studies have focused on alternatives to conventional antimicrobials. It is known that the environment is a key source of hospital pathogens [2], therefore attention has been focussed on the disinfection of hospitals, in particular using antibacterial treatments carried out in novel ways, including antibacterial light sources [3,4,5]. Such new strategies, capable of decontaminating both patient wounds and the environment, are important tools for the fight against dangerous hospital pathogens. The best-known strategy that uses light, is photodynamic therapy (PDT), in which light, a photosensitizer, and oxygen kill undesired cells. It is well established for the treatment of several cancers, and has been successfully employed for the treatment of skin tumours [5,6], cutaneous T-cell lymphoma [7], and tumors localized in the oral cavity and tongue [8,9]. In addition, precancerous lesions, such as Bowen’s disease, early stages of cervical cancer, and Barrett’s oesophagus can be treated with PDT [10].

There is now growing interest in antimicrobial PDT (aPDT), in which pathogenic microorganisms are inactivated [11]. In aPDT, a photosensitizer is activated with visible light of an appropriate wavelength(s) to generate toxic species that inactivate the microorganisms [12]. One of the best-known photosensitizers is methylene blue (MB). Photosensitization reactions induced by MB excitation are known to cause damage to several biomolecules. This damage is thought to be triggered both by type I and type II processes. In the type I mechanism, free radicals can react with proteins and/or lipids, leading to a chain reaction that produces more oxidation products. In the type II mechanism, energy from the triplet state of the photosensitizer is transferred to molecular oxygen, resulting in the generation of highly reactive singlet oxygen. The singlet oxygen can directly react with cellular molecules in its immediate vicinity, and also creates further oxygen radicals. Singlet oxygen and several other reactive species involving oxygen (e.g., the OH radical) are collectively called reactive oxygen species (ROS) [13].

It has previously been found that differences in the cell wall structure mean that Gram-negative bacteria are harder to treat by PDT [14]. Gram-positive species are more susceptible to photodynamic inactivation because their cell wall, located outside the cytoplasmic membrane, is a relatively porous structure that is permeable to molecules with a molecular weight in the 30,000–60,000 Da range, and so it allows photosensitizers to cross it [15]. The complex structure of the cell wall makes Gram-negative bacteria poorly permeable to many photosensitizers and photogenerated reactive species [14]. The porin channels located in the outer, negatively charged membrane, can allow penetration of some photosensitizers, but these need to be cationic, hydrophilic compounds with low molecular weight (less than 700 Da) to achieve good penetration inside the cell membrane or cytoplasm [16].

Among the many solutions that can improve the efficiency of aPDT against Gram-negative bacteria, there is great interest in the use of photosensitizers in combination with metal nanoparticles, because nanoparticles have been shown to be able to pass through the cell walls of both Gram-negative and Gram-positive bacteria [17]. Currently, several options for photosensitizers are present in the literature, and it has been shown that their combination with metallic nanoparticles is a promising strategy for improving antimicrobial activity [18,19,20,21].

In this paper we show how biogenic gold nanoparticles can lead to effective aPDT of both Gram-negative (*Escherichia coli*) and Gram-positive (*Staphylococcus aureus*) bacteria. Biogenic nanoparticles are nanoparticles made by a living organism. Here we use gold nanoparticles made by fungi in a simple, one-step, and environmentally friendly synthesis. The biogenic synthesis enables control of the shape and size distribution, and does not require external stabilising agents, because the nanoparticles are stabilised by components of the biological reaction. In comparison to chemically synthesized nanostructures, the gold nanoparticles produced by fungi are free from the toxic contamination of by-products that become attached to the nanoparticles during chemical synthesis, and which have so far limited the use of the resulting gold nanoparticles in biomedical applications [22]. This leads to the key advantage of the biologically produced nanoparticles for our application, better biocompatibility [23], and this in turn leads to greater effectiveness for PDT, and so enables us to use a compact and simple LED based light source. A further advantage is that biogenic gold nanoparticles have very good stability [24].

## 2. Results

### 2.1. Synthesis and Characterization of the Biogenic Gold Nanoparticles

The extinction spectrum of the obtained nanoparticles was characterized by UV–visible spectroscopy, and as shown in Figure 1a, the absorption of the biosynthesized gold nanoparticles showed a single plasmon band at 522–523 nm. Figure 1b shows a representative TEM micrograph recorded from drop-coated films of the gold nanoparticles synthesized by the mycelium of *Mucor plumbeus.*

These particles are spherical in shape and uniformly distributed, without any significant agglomeration. The particle size histogram (Figure 1c) shows that the nanoparticle sizes ranged from 13 to 25 nm and possessed an average size of 17 ± 4 nm. The frequency of distribution observed from the histogram indicates that almost 77% of the gold particles were in the 19- to 22-nm size range.

### 2.2. Photo-Inactivation of Bacteria

The study of the photo-inactivation of *S. aureus* and *E. coli* started by determining the toxicity of methylene blue in the dark. Table 1 shows the effect of MB at concentrations ranging from 6.25 mgL^−1^ to 250 mgL^−1^ on the viability of *S. aureus* and *E. coli*. As can be seen, the dark effect of the dye increased with the dye concentration (Table 1). The values for the reduction in viability of planktonic cells of *S. aureus* were 61 ± 3, 45 ± 4, 28 ± 4, 18 ± 3, 11 ± 2, and 8 ± 2% for MB concentrations of 250, 100, 50.0, 25, 12.5, and 6.25 mgL^−1^, respectively. In the case of *E. coli* the reduction in viability was 45 ± 4, 15 ± 3, 9 ± 2, 7 ± 2, 5 ± 1, and 3 ± 1% for MB concentrations of 250, 100, 50.0, 25, 12.5, and 6.25 mgL^−1^.

The biogenic gold nanoparticles (AuNPs) up to the concentration of 20 ppm (in the dark) did not affect the number of bacteria cells (data not shown). Changes in the value of colony-forming units (CFU)/mL were insignificant, and were within the measurement error. The dark toxicity studies on the MB + AuNPs system (the final concentration of MB was 12.5 mgL^−1^ and 100 mgL^−1^ for *S. aureus* and *E. coli*, respectively) showed that these mixtures have a higher antibacterial activity than MB alone, but the unit of reduction in viability of *S. aureus* and *E. coli* was not higher than 22 ± 3%.

Experiments on the photo-inactivation of *S. aureus* and *E. coli* were performed for three different light intensities (2.5 cm^−2^, 5 mW cm^−2^, and 10 mW cm^−2^), and the results are presented below, in order of increasing light intensity. For each light intensity, three durations of illumination were used, and cell viability was measured in: a control sample (light, but no methylene blue or nanoparticles); a sample with MB but no light; a sample with nanoparticles but no light; a sample with MB and light; and a sample with MB, nanoparticles and light. For each set of conditions, we present the results for *S. aureus* followed by those for *E. coli*.

The results for the lowest light intensity of 2.5 mW/cm^2^ are shown in Figure 2a,b. As can be seen, the AuNPs showed an insignificant 0.22, 0.27, and 0.30 log_10_ unit reduction in *S. aureus* culture viability after irradiation with energy fluences of 0.75, 2.25, and 4.5 J cm^−2^, respectively (Figure 2a; see also Table 2). When MB alone was used as photosensitizer the highest mean reduction in the number of living cells was after 30 min of irradiation and was 2.12 log_10_, that is 99.25% kill. Shorter times of light treatment resulted in a slight decrease in the number of bacterial cells, and the reduction in CFU was 0.6 log_10_ and 0.57 log_10_, after 5 and 15 min, respectively. The MB + AuNPs mixture showed a significant 1.3 log_10_ and 1.6 log_10_ unit reduction in *S*. *aureus* culture after 5 and 15 min of irradiation, that is 95% and 97.5%. The most effective reduction in the number of cells was found for MB + AuNPs after 30 min of LED light irradiation (4.5 J cm^−2^), and was 2.88 log_10_, that is 99.87% kill, compared with 1.12 log_10_ or 99.25% for the same illumination time with MB alone (see Table 2).

When an LED output of 2.5 mW cm^−2^ was used for photo-inactivation of *E. coli*, the AuNPs showed 0.23 log_10_ (53% kill), 0.25 log_10_ (55% kill), and 0.3 log_10_ unit (60% kill) reduction in *E. coli* culture viability after irradiation with energy fluences of 0.75, 2.25, and 4.5 J cm^−2^, respectively (Figure 2b; see also Table 2). The mortality achieved by MB as a photosensitizer was light dose-dependent, with the kill increasing as the exposure time was increased from 5 to 30 min. The MB had photobactericidal activity, and a reduction of 0.33and 0.41 log_10_ unit in *E.coli* culture after 5 and 15 min of irradiation was observed. The most effective reduction in the number of cells with MB as photosensitizer was found after 30 min of LED light irradiation, and was 1.77 log_10_, that is 98.65% kill. When the MB + AuNPs mixture was used in experiments, after 5, 15, and 30 min of diode irradiation the viable count showed a reduction of 0.98 log_10_ (89.55% kill), 1.82 log_10_ (94.3%), and 2.0 log_10_ (99.0%), respectively. It should be noticed that the photosensitization of *S. aureus* and *E. coli* cells was synergistically enhanced by the biological gold nanoparticles.

We continued our study using a higher light output of 5 mW cm^−2^. The AuNPs showed an insignificant 0.24, 0.27, 0.32, and 0.34 log_10_ unit reduction in *S. aureus* culture viability after irradiation with energy fluences of 1.5, 4.5, 9, and 13.5 J cm^−2^, respectively (Figure 3a; see also Table 2). At the same conditions, a significant destruction (from 0.63 to 3.27 log_10_ unit reduction) of *S. aureus* cells was achieved when MB was used as a photosensitizer. Photoinactivation of the studied coccus with MB was time dependent, and the highest mortality was observed after 45 min of irradiation with an energy fluence of 13.5 J cm^−2^ (99.947% kill). The MB + AuNPs mixture appeared to be the most active photosensitizer, as a modest dose of 13.5 J cm^−2^ resulted in a 3.6 log_10_ unit reduction of viability, that is 99.978% kill.

When 5 mW cm^−2^ was used to kill *E. coli*, the AuNPs revealed 0.34, 0.37, 0.41, 0.44, and 0.47 log_10_ unit reduction after 5, 15, 30, and 45 min of irradiation, respectively (Figure 3b; see also Table 2). Appreciable destruction of cells of *E. coli* was achieved using MB as a photosensitizer. The MB showed a 0.43 and 1.78 log_10_ unit reduction in *E. coli* planktonic cells after 5 and 15 min of irradiation. A longer exposure time (30–45 min) resulted in a reduction in CFU of 2.20 and 3.60 log_10_, that is 99.375% and 99.975% kill. The most effective reduction in the number of cells was found after 60 min of laser light irradiation, and was 3.92 log_10_, that is 99.988% kill. A mixture of MB + AuNPs greatly enhanced the photoinactivation of *E. coli* as 5, 15, and 30 min irradiation resulted in 1.19, 1.96, and 2.37 log_10_ units of viability reduction, respectively.

In the third set of our experiments, the LED array was set to its highest light intensity (10 mW cm^−2^). The largest reduction in the number of *S. aureus* with the AuNPs as photosensitizer was observed after 45 min of irradiation (27 J cm^−2^), and was 0.37 log_10_, that is 57.38% kill (Figure 4a; see also Table 2). It can be seen that after 5, 15, 30, and 45 min of light irradiation, which corresponded to an energy fluence of 3, 9, 18, and 27 J cm^−2^, in the presence of MB as photosensitizer, the viable count showed a reduction of 1.22, 3.08, 3.30, and 5.2 log_10_, respectively. A very high reduction in the number of live cells (2.18 log_10_ unit, 99.344% kill) was observed after 5 min of exposure to light irradiation with the MB + AuNPs mixture as photosensitizer. A longer exposure time (15 and 30 min) resulted in a reduction in CFU of 3.25 log_10_ (99.94% kill) and 3.34 log_10_ (99.96% kill), respectively. It should be noted that in this case, a clear relationship between exposure time and cell mortality was also not observed. Hence, the combination of MB with AuNPs greatly enhanced photoinactivation of *S. aureus,* as 45 min irradiation resulted in a lethal effect (the number of bacteria was below the detection level).

When this LED was used in photosensitization of *E. coli*, the mortality achieved by the AuNPs was light dose-dependent, with the kill increasing as the exposure time was increased from 5 to 60 min (Figure 4b; see also Table 2). It was found that after 5, 15, and 30 min of light treatment the viable count showed a reduction of 0.37 log_10_ (57% kill), 0.42 log_10_ (62% kill), and 0.47 log_10_ (66.5% kill), respectively. Prolongation of expose time to 45 and 60 min caused a further slight increase in the mortality of bacterial cells to 0.49 log_10_ (68% kill) and 0.52 log_10_ (70% kill). After 5, 15, and 30 min of light treatment with MB as photosensitizer, a reduction of 1.04, 2.30, and 2.6 log_10_ was obtained, respectively. The higher mean reductions in the number of bacterial cells were observed after 45 and 60 min of irradiation, and were 3.82 and 3.95 log_10,_ respectively (that is 99.985% and 99.986% kill). A very effective reduction in the number of cells was found when a MB + AuNPs mixture was used as a photosensitizer, and after 5 and 15 min of irradiation the viable count showed a reduction of 1.67 and 2.89 log_10_, that is 97.89% and 99.87% kill. A longer exposure time (30, 45 min) resulted in a reduction in CFU of 3.04 and 3.85 log_10,_ that is 99.91 and 99.986% kill, respectively. The most effective reduction in the number of cells was found after 60 min of LED light irradiation (36 J cm^−2^) and was 4 log_10_, that is 99.99% kill, compared with 3.95 log_10_ or 99.986% for MB. Hence, the combination of MB with the AuNPs was again most effective.

### 2.3. Photobleaching of Methylene Blue

Experiments on the influence of the LED light on the solution of the MB in water, and on the solution consisting of the MB + AuNPs, were performed, and the results are presented on the Figure 5a,b, which show the variation in the absorption spectra of methylene mlue aqueous solution and MB + AuNPs under irradiation with LED light, measured at intervals of 4 min.

It was observed that the characteristic absorption peak of MB around 662 nm decreased gradually, becoming broadened with the increasing of irradiation time, and shifted slightly toward a shorter wavelength. It should be noticed that the characteristic absorption peak of the biogenic gold nanoparticles at 521 nm was unchanged during the irradiation time (Figure 5b). The presence of AuNPs in the photosensitizing mixture inhibited the fast photobleaching of MB.

Figure 6 shows a kinetic profile of MB photobleaching in the absence (black balls) and presence (opened squares) of the biogenic gold nanoparticles (AuNPs).

As can be clearly seen from Figure 6, the change of the concentration of the MB upon photobleaching of the MB without gold nanoparticles was much faster than the photobleaching of the mixture of MB with the gold nanoparticles. The process of photofading was less effective in the MB + AuNPs system than in the solution of MB, thus enhancing the effectiveness of the singlet oxygen production in the mixture, in comparison to the MB alone.

## 3. Discussion

Under the conditions defined in our studies we found that antimicrobial PDT in vitro reduces the concentration of *Staphylococcus aureus* and *Escherichia coli* viable cells, and that this process is enhanced by biogenic gold nanoparticles.

Methylene blue was chosen as the photosensitizer because it is a well-known phenothiazine dye used in medicine as a therapeutic agent [21] or a photosensitizing compound [22]. This molecule is of particular interest for use in PDT due to its physicochemical properties. MB exhibits a quantum yield of singlet oxygen formation of about 0.5, with low reduction potential, and intense light absorption between 550 nm and 700 nm in water (in the phototherapeutic window). It is well known that MB is a highly hydrophobic compound, with a higher chemical affinity to nucleic acids, and showing low levels of toxicity in mammalian cells [25]. Currently, MB is used by several European agencies for the disinfection of blood plasma, due to its effectiveness in the photodynamic inactivation of microorganisms, such as bacteria [26], and viruses [27], including HIV, and hepatitis B and C [28,29,30,31]. In addition, an important point to consider is the much lower cost of treatment based on this dye, compared to other available photosensitizers.

It was previously shown that MB has a significant antimicrobial effect in the dark, and that this activity can be increased in the presence of oxygen by applying light with a wavelength corresponding to its electronic absorption band [30]. Therefore, it was reasonable to study the dark bactericidal activity of this dye. From the results collected in Table 2, it follows that the reduction in number of *S. aureus* cells was insignificant up to a MB concentration of 12.5 mgL^−1^. In the case of *E. coli,* insignificant reductions in the number of cells were observed up to a concentration of 100 mgL^−1^ of MB. These observations show that the dark toxicity of MB depends on the type of bacteria: Gram-positive coccus was more sensitive to MB than Gram-negative rods. These findings are consistent with the results obtained by Usacheva et al. [32], who hypothesized that this is due to the presence of an outer membrane in Gram-negative bacteria.

The detailed analysis of the results seen in Figure 2, Figure 3 and Figure 4 indicates that, regardless of the radiation power density used, the biogenic gold nanoparticles synthesized by *M. plumbeus* synergistically enhanced the kill of the bacteria studied, whilst showing insignificant antibacterial activity. The degree of photodamage was dependent upon the light fluence and the intensity of the LED light, as well as the genus of bacteria. The relationship between energy fluence (J cm^−2^) and bacterial cell mortality rate (%) is summarized in Table 2. The table shows that energy fluences in the range 0.75–1.5 J cm^−2^, with MB as photosensitizer caused only limited mortality of *S. aureus,* of up to 77%. In the same experimental conditions, the combination of MB and AuNPs was more efficient, and the mortality rate of coccus was significantly higher, and exceeded 97%. The use of a MB + AuNPs mixture as a photosensitizer reduced the energy fluence needed to kill over 99% of *S. aureus,* to 4.5 J cm^−2^ compared to MB alone, where an energy fluence of 9 J cm^−2^ was required. In addition, an energy fluence of 27 J cm^−2^ and MB + AuNPs mixture resulted in a lethal effect on *S. aureus* (the number of cells was below the detection level). The American Society of Microbiology has decreed that any antimicrobial technique is required to kill a minimum of 3 logs of CFU (99.9%) in order to be accepted as “antimicrobial”.

Energy fluences in the range 0.75–1.5 J cm^−2^ and MB as photosensitizer caused more than 71% mortality of *E. coli* cells. In the same experimental conditions, a MB + AuNPs mixture was more effective, and the mortality rate of the studied rods was significantly higher, and exceeded 95%. It was shown that mortality of *E. coli*, regardless of a photosensitizer, was achieved at the level of 99.9%, with an energy fluence of 13.5 J cm^−2^. However, the mortality rate was higher with the MB + AuNPs mixture, and reached 99.987%. The highest photobactericidal activity against *E. coli* (99.99%) was observed with a MB + AuNPs mixture as photosensitizer, with an energy fluence of 36 J cm^−2^.

Our results show that the use of biogenic gold nanoparticles to improve the effectiveness of antibacterial PDT is an attractive approach. Previously, it was demonstrated that a higher efficiency of photobactericidal effect can be achieved in various ways, e.g., encapsulation of the photosensitizer (PS) in nanoparticles, increasing PS delivery to microorganisms, or increasing PS yield by covalently bonding PS to the surface of nanoparticles [33]. It is worth noting that those studies were carried out under different conditions, e.g., different concentrations of photosensitizer, time of exposure, and energy dose of light, etc., and therefore a direct comparison cannot be easily made.

The mechanism responsible for enhancement of the photo-bactericidal effect by gold nanoparticles has been discussed in the literature [34,35,36]. Most often it is suggested that the presence of AuNPs changes the relative distribution of ROS agents or increases their production. Narband et al. [37] indicated that gold nanoparticles enhance the bacterial kill by encouraging less dye fluorescence and formation of excited oxygen species other than singlet oxygen. Our work shows that the combination of methylene blue and the AuNPs slows down the process of photobleaching of MB by one order of magnitude, in comparison to the solution of the dye alone.

Photobleaching of MB has been studied in some detail, and the role of the triplet excited state has been recognized [38,39]. This dye fades relatively quickly when exposed to light, even in the absence of electron donors, and has been reported to retain a two-step process, involving intermediary formation of singlet oxygen, on exposure to visible light in aqueous solution [40].

The leuco form of MB, produced by adding a hydride anion, is most often mentioned as a candidate for a transparent product, and the relationship between the highly-colored oxidized form of methylene blue (MB^+^), and its colourless reduced leuco form (LMB) is presented in Figure 7 [41,42,43].

At low in vivo concentrations, methylene blue and leuco-methylene blue are in equilibrium, such that they form a reversible reduction/oxidation system.

One of the possible mechanisms limiting the photobleaching of a chromophore is their placement in the vicinity of a metallic surface, such as f.i. gold nanoparticles. The overall fluorescence lifetime decreases as the dye molecule approaches the gold nanoparticle surface [44,45,46], which typically increases the stability of the chromophore due to reduction of the probability of the excited state molecule reacting with the nearby oxygen that leads to photobleaching [47,48,49]. It was also previously pointed out that the deactivation process of the excited state of chromophores, through interactions with the metallic surface, enhanced their photochemical stability [50].

Here we show clearly, the effect of the presence of the gold nanoparticles on the photobleaching of MB, which is manifested by the change of the kinetics of the photofading process. The gold nanoparticles prevent fast photobleaching, thus enhancing the photoactivity of the MB molecules and improving their bactericidal effect. It cannot be excluded that other mechanisms, e.g., a local increase in the concentration of the photosensitizer through targeted delivery of nanoparticles, selective interaction with the bacterial cell wall, or resonant heating of AuNP in irradiation with laser light, could also be involved in the more efficient killing of bacteria [51].

## 4. Materials and Methods

### 4.1. Reagents

All chemicals agents, including tetrachloroauric acid and methylene blue were obtained from POCH, Poland. Methylene blue solution (MB) was prepared by dissolving the powdered dye in deionized water, and sterilized by filtration through 0.22-µm pore diameter membranes (Millex^®^-HP syringe-driven filter unit, Millipore, Warszawa, Poland). After filtration, the photosensitizer solution was stored in the dark.

### 4.2. Light Source

We built a light source for PDT consisting of 9 LEDs (Kingbright KA-3529ASEL2Z4S, Northants, UK) with a peak wavelength of 640 nm and FWMH 20 nm, arranged in a 3 × 3 array. The light source was driven to give a user-selected calibrated output of 2.5 mW cm^−2^, 5 mW cm^−2^, or 10 mW cm^−2^. The light source was placed on the top of a microwell plate, and the stated output of the LEDs was calibrated to halfway down the wells as shown at Figure 8.

### 4.3. Synthesis and Characterization of Gold Nanoparticles

The gold nanoparticles were synthesized by the fungus *Mucor plumbeus* according to the procedure described previously, with some modifications [52]. In detail, the fungus was incubated with gold ions at 35 °C for 18 h. To isolate the produced gold nanoparticles, the fungi were washed twice with deionized water, and ultrasonic disruption of cells was carried out with an ultrasonic processor (TURBO 36800, Polsonic, Poland) over five-six 10 s periods, and with an interval of 60 s between periods. Then the sonicated samples were centrifuged at 3500 rpm for 20 min at 4 °C to remove the cell debris. The obtained gold nanoparticles were separated by the sucrose density gradient technique described by Maliszewska [53], and spheres concentrated in the 40% fraction were studied. To verify the reduction of gold ions, the extinction spectra of the solutions were recorded in the range of 200–800 nm with a spectrophotometer (UV-1650 PC, Shimadzu, Japan). The size and morphology of the nanoparticles were analyzed with a transmission electron microscope (Zeiss EM 900, Oberkochen, Germany). The sample was prepared by placing a drop of metallic nanoparticles on a carbon-coated copper grid, and subsequently drying in air before transferring it to the microscope. The particle size distribution was determined from electron micrographs of at least 300 particles.

### 4.4. Dark Toxicity Assays

The reference strains *S. aureus* PCM 2054 and *E. coli* PCM 2058 were seeded onto Mueller Hinton agar (BIOCORP, Warsaw, Poland) and incubated at 37 °C for 24 h. After that period, the bacteria were cultured in Mueller Hinton broth (BIOCORP, Poland) at 37 °C for an additional 20–22 h. Then, 2 mL of each culture was centrifuged at 1300× *g* for 10 min and washed twice in PBS. The cell pellet was re-suspended in phosphate buffered saline (PBS) to give an inoculum of approximately 1.6 × 10^6^ colony-forming units (CFU mL^−1^) for *S.*
*aureus,* and 2.0 × 10^6^ colony-forming units (CFU mL^−1^) for *E. coli*, respectively. Sterile MB solutions with concentrations of 750, 300, 150, 75, 37.5, and 18.75 mgL^−1^ were also prepared. Then, to each well of a 96 well flat-bottomed microwell plate (FL Medical, Italy) 100 µL of standardized suspension of the test strains and 50 µL of the appropriate MB concentration were added to obtain final MB concentrations of 250, 100, 50, 25, 12.5, and 6.25 mgL^−1^. Suspensions containing the same concentration of microorganisms and different dye concentrations were incubated at 37 °C, with shaking for 60 min (in the dark). Then, the suspensions were serially diluted in saline, and 100 µL aliquots were spread over the Mueller Hinton agar surface. Plates were then incubated for 24 h at 37 °C and examined for colony forming units per milliliter. All bacterial and media manipulations were performed with minimal exposure to ambient light. To evaluate antibacterial activity, the percentage reduction (%) of bacteria was calculated as the reduction in viability R = (N_0_ − N) × 100/N_0_, where N_0_ and N are the numbers of CFUs at initial (1.6 × 10^6^ CFU mL^−1^ for *S.*
*aureus* and 2.0 × 10^6^ CFU mL^−1^ for *E. coli*), and remaining in suspension after incubation with MB under dark conditions. The culture of the studied microorganism was incubated under the same conditions, and was used as a control. The optimal conditions were determined to be the highest MB concentrations that resulted in no more than 15% cell mortality without exposure to light.

The effect of the biogenic AuNPs and MB + AuNPs mixture on the growth of *S. aureus* and *E. coli* was studied under the same conditions as described above. The concentration of AuNPs was 20 ppm.

### 4.5. In Vitro Photodynamic Inactivation of Bacteria

The bactericidal effect of aPDT was examined in a sterile 96-well flat-bottom microtiter plate (F.L. Medical, Florence, Italy). Briefly, 100 µL aliquot of the standardized suspension of the tested strain was added to each well of a 96-well. Then, the assays were divided into six experimental groups: treatment with gold nanoparticles only (AuNPs-L, *n* = 3); treatment with gold nanoparticles and LED irradiation (AuNPs + L, *n* = 3) (the assay groups AuNPs-L and AuNPs + L received 100 µL of the biogenic gold nanoparticles); treatment with MB at the concentration of 12.5 mgL^−1^ or 100 mgL^−1^ only (MB-L, *n* = 3); treatment with MB at the concentration of 12.5 mgL^−1^ or 100 mgL^−1^ and LED irradiation (MB + L, *n* = 3) (the assay groups MB-L and MB + L received 100 µL of MB); treatment with gold nanoparticles and MB at the concentration of 12.5 mgL^−1^ or 100 mgL^−1^ only (AuNPs + MB-L, *n* = 3); and treatment with gold nanoparticles and MB at the concentration of 12.5 mgL^−1^ or 100 mgL^−1^ and LED irradiation (AuNPs + MB + L, *n* = 3) (the assay groups AuNPs + MB-L and AuNPs + MB + L received 100 µL of MB-gold nanoparticles mixture).

The plate was then shaken for 20 min (pre-irradiation) in an orbital shaker. The wells containing the assay groups +L were then exposed to LED light for various periods of time (5 min, 15 min, 30 min, 45 min, and 60 min). Twelve additional wells containing the bacterial suspension (100 µL) and PBS (100 µL) were prepared. Six of these were exposed to LED light to determine the effect of light alone on bacterial viability. The remaining six were kept in the dark as an overall control, and to determine the initial concentration of bacteria in the suspensions. After irradiation or incubation in the dark, samples were serially diluted and spread in duplicate onto Mueller Hinton agar plates. The plates were then incubated aerobically at 37 °C for approximately 24 h. After incubation, the number of bacteria surviving each treatment was determined and the concentration of survivors expressed as CFU mL^−1^.

The effectiveness of photodynamic reduction in bacterial viability was assessed as: (1) reduction in CFUs (log_10_ CFUml^−1^) calculated as R = log_10_N_0_ − log_10_N, and (2) percent reduction (%) in bacterial viability calculated as R = (N_0_ − N) × 100/N_0_, where N_0_ and N are the numbers of CFUs at the beginning (1.6 × 10^6^ CFU ml^−1^ for *S. aureus* and 2.0 × 10^6^ CFU mL^−1^ for *E*. *coli*, respectively) and remaining in suspension after photoinactivation.

### 4.6. Photobleaching of MB

The methylene blue and mixture of MB + AuNPs were irradiated for 60 min. After each 4 min, a UV spectrum of MB and MB + AuNPs in the range of 400–800 nm was generated. The UV spectra and the kinetic curves of photobleaching of MB and MB + AuNPs were determined.

### 4.7. Statistical Analysis

All experiments were run in triplicate, i.e., 3 biological repetitions, each with 3 technical repetitions. The statistical analyses were performed using STATISTICA data analysis software (version 10.0) and Excel. The quantitative variables were characterized by the arithmetic mean, standard deviation, median, max/min (range), and 95% confidence interval. The statistical significance of differences between two groups was processed with the Student’s *t* test. In all the calculations, a *p*-value of 0.05 was used as the cut-off for statistical significance.

## 5. Conclusions

We have demonstrated that biogenic gold nanoparticles are effective for aPDT. In particular we showed that in combination with methylene blue, they can lead to effective killing of both Gram-negative and Gram-positive bacteria. The enhanced effectiveness that we demonstrated means that lower light intensities can be used. This is important as it enables the use of compact and potentially wearable LED-based light sources. Hence our results provide a pathway to simple, low cost treatment of superficial infections by both Gram-positive and Gram-negative bacteria.

## Figures and Tables

**Figure 1 molecules-26-00623-f001:**
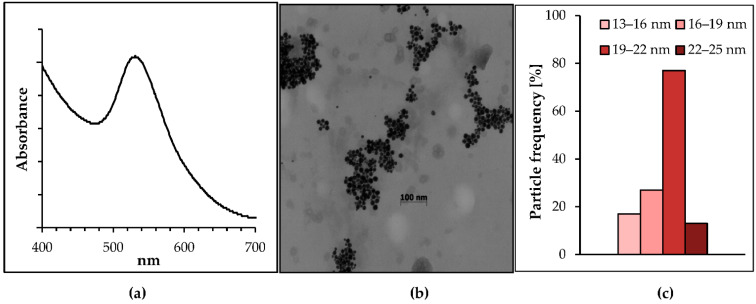
(**a**) Absorption spectrum of the biogenic gold nanoparticles; (**b**) TEM micrographs of the gold nanoparticles (**b**); (**c**) the particle size histogram of the gold nanoparticles.

**Figure 2 molecules-26-00623-f002:**
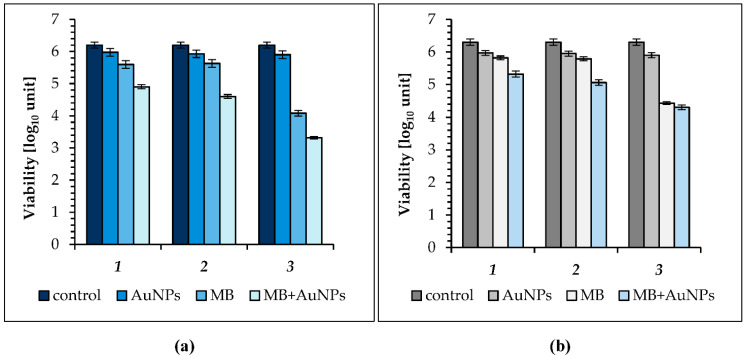
(**a**) Effect of AuNPs (gold nanoparticles), MB (methylene blue) and the MB + AuNP (methylene blue-gold nanoparticle mixture) on viability of: *S. aureus* following incubation in the dark: (**1**) exposure to LED light for 5 min (energy fluence was 0.75 J cm^−2^); (**2**) 15 min (energy fluence was 2.25 J cm^−2^); (**3**) 30 min (energy fluence was 4.5 J cm^−2^). (**b**) *E. coli* following incubation in the dark: (**1**) exposure to LED light for 5 min (energy fluence was 0.75 J cm^−2^); (**2**) 15 min (energy fluence was 2.25 J cm ^−2^); (**3**) 30 min (energy fluence was 4.5 J cm^−2^); (in all groups *p* < 0.05); the control is the initial concentration of bacteria in suspensions kept in the dark.

**Figure 3 molecules-26-00623-f003:**
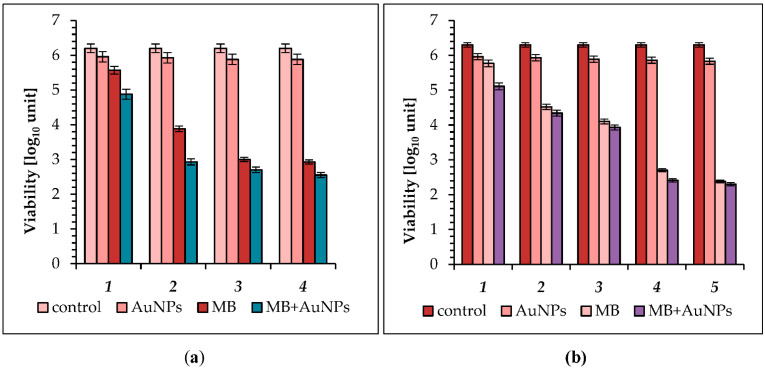
(**a**) Effect of AuNPs (gold nanoparticles), MB (methylene blue) and the MB + AuNP (methylene blue-gold nanoparticle mixture) on viability of: *S. aureus* following incubation in the dark: (**1**) exposure to LED light for 5 min (energy fluence was 1.5 J cm^−2^); (**2**) 15 min (energy fluence was 4.5 J cm^−2^); (**3**) 30 min (energy fluence was 9 J cm^−2^); (**4**) and 45 min (energy fluence was 13.5 J cm^−2^). (**b**) *E. coli* following incubation in the dark: (**1**) exposure to LED light for 5 min (energy fluence was 1.5 J cm^−2^); (**2**) 15 min (energy fluence was 4.5 J cm^−2^); (**3**) 30 min (energy fluence was 9 J cm ^−2^); (**4**) 45 min (energy fluence was 13.5 J cm^−2^); and (**5**) 60 min (energy fluence was 18 J cm^−2^); (in all groups *p* < 0.05); the control is the initial concentration of bacteria in suspensions kept in the dark.

**Figure 4 molecules-26-00623-f004:**
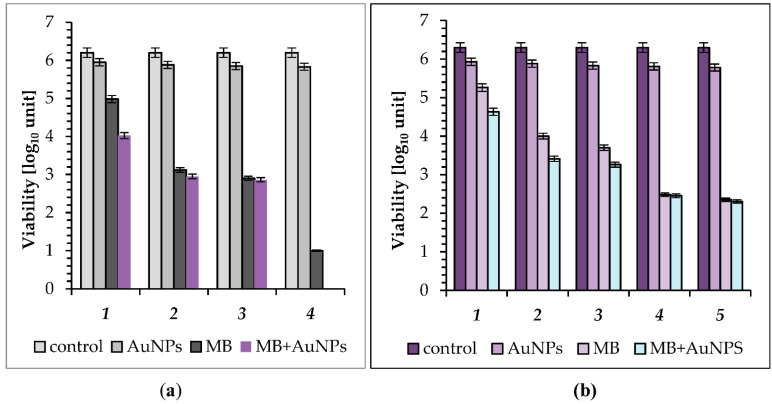
(**a**) Effect of AuNPs (gold nanoparticles), MB (methylene blue), and the MB + AuNP (methylene blue-gold nanoparticle mixture) on viability of: *S. aureus* following incubation in the dark: (***1***) exposure to LED light for 5 min (energy fluence was 3 J cm^−2^); (***2***) 15 min (energy fluence was 9 J cm^−2^); (***3***) 30 min (energy fluence was 18 J cm^−2^); (***4***) and 45 min (energy fluence was 27 J cm^−2^). (**b**) *E. coli* following incubation in the dark: (***1***) exposure to LED light for 5 min (energy fluence was 3 J cm^−2^); (***2***) 15 min (energy fluence was 9 J cm^−2^); (***3***) 30 min (energy fluence was 18 J cm^−2^); (***4***) 45 min (energy fluence was 27 J cm^−2^); and (***5***) 60 min (energy fluence was 36 J cm^−2^); (in all groups *p* < 0.05); the control is the initial concentration of bacteria in suspensions kept in the dark.

**Figure 5 molecules-26-00623-f005:**
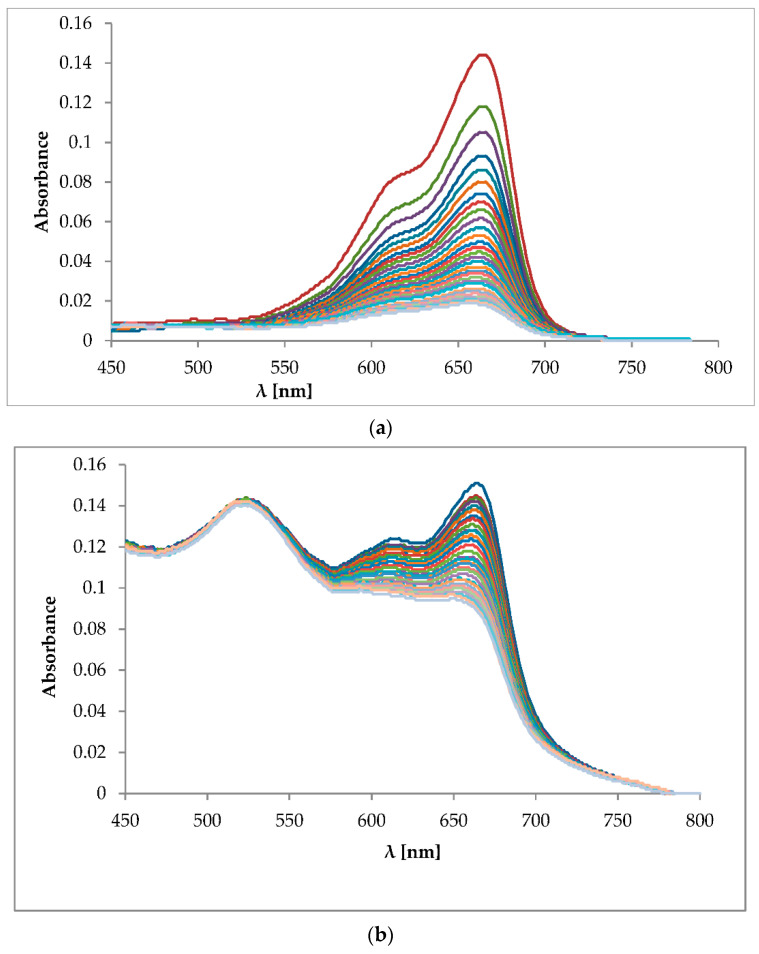
(**a**) Variation in the optical absorption spectra of MB during LED light irradiation (640 nm) for 60 min. The spectra were registered with a 4 min interval, the first one (highest absorbance at 662 nm) is non-irradiated solution of MB. (**b**) Variation in the optical absorption spectra of MB + AuNPs during LED light irradiation (640 nm) for 60 min. The spectra were registered with a 4 min interval, the first one, with the highest absorbance at 662 nm, shows absorbance of the solution of MB + AuNPs without irradiation

**Figure 6 molecules-26-00623-f006:**
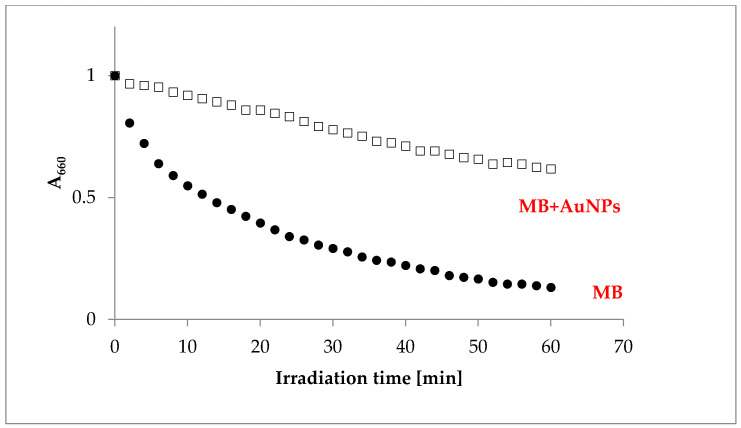
Kinetic curves of MB photobleaching in the absence (●) or presence (□) of AuNPs.

**Figure 7 molecules-26-00623-f007:**
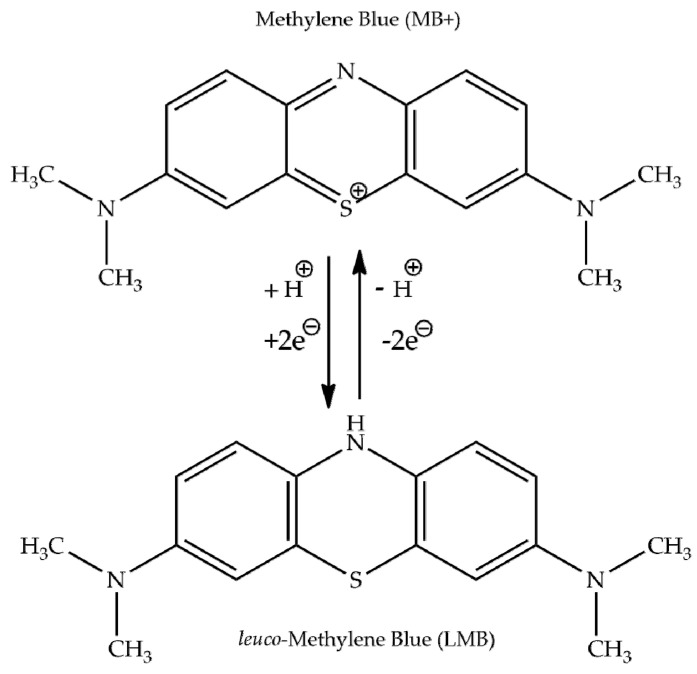
Molecular formulae for methylene blue and its leuco-form.

**Figure 8 molecules-26-00623-f008:**
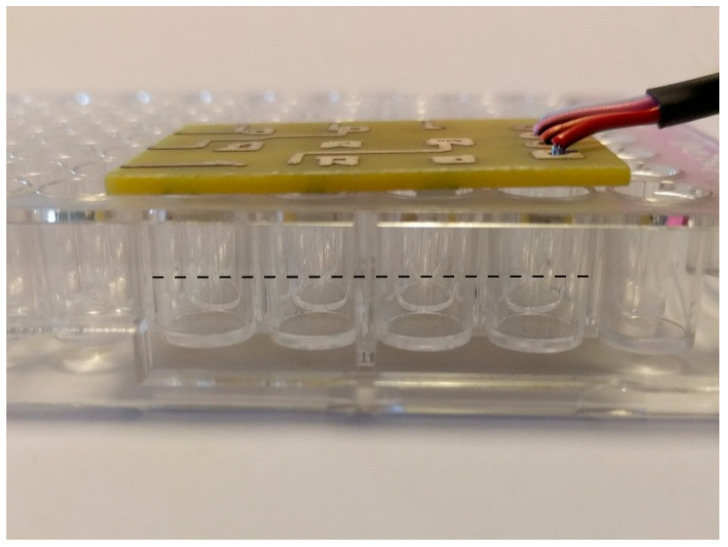
Photograph showing the LED array above microwell plate.

**Table 1 molecules-26-00623-t001:** Bactericidal efficacy of methylene blue (MB) against *Staphylococcus aureus* and *Escherichia coli* under dark conditions.

Concentration of Methylene Blue [mgL^−1^]	*S. aureus*	*E. coli*
Values of reduction in viability [%] ^1^		
6.25	8 ± 2	3 ± 1
12.5	11 ± 2	5 ± 1
25.0	18 ± 3	7 ± 1
50.0	28 ± 4	9 ± 2
100.0	45 ± 4	15 ± 3
250.0	61 ± 3	45 ± 4

^1^ Mean values of three replicates ± standard deviation of mean.

**Table 2 molecules-26-00623-t002:** The relationship between energy fluence (J cm^−2^)/power density and bacterial cell mortality rate (%).

Energy Fluence (J cm ^−2^)/Power Density	Mortality Rate (%)					
	*S. aureus*			*E. coli*		
	AuNPs	MB	MB + AuNPs	AuNPs	MB	MB + AuNPs
0.75/2.5 mW cm^−2^	40.3 ± 0.1	75.0 ± 0.3	95.0 ± 0.2	53 ± 1	67.25 ± 0.05	89.55 ± 0.04
1.5/5 mW cm^−2^	43 ± 1	76.56 ± 0.05	95.312 ± 0.008	54.4 ± 0.3	70.5 ± 0.4	93.5 ± 0.2
2.25/2.5 mW cm^−2^	46.9 ± 0.1	73.125 ± 0.005	97.5 ± 0.3	55 ± 2	69.0 ± 0.3	94.3 ± 0.1
3/10 mW cm^−2^	44.4 ± 0.2	94.063 ± 0.007	99.344 ± 0.007	57 ± 3	91.0 ± 0.5	97.89 ± 0.03
4.5/2.5 mW cm^−2^	50.4 ± 0.3	99.25 ± 0.04	99.869 ± 0.008	60 ± 3	98.65 ± 0.03	99.0 ± 0.3
4.5/5 mW cm^−2^	46.9 ± 0.2	99.532 ± 0.007	99.947 ± 0.007	57.4 ± 0.4	98.35 ± 0.04	98.9 ± 0.5
9/5 mW cm^−2^	52.5 ± 0.3	99.938 ± 0.006	99.969 ± 0.009	61.2 ± 0.3	99.375 ± 0.005	99.575 ± 0.004
9/10 mW cm^−2^	52.5 ± 0.3	99.918 ± 0.007	99.94 ± 0.04	62 ± 1	99.5 ± 0.4	99.87 ± 0.03
13.5/5 mW cm^−2^	54.7 ± 0.2	99.947 ± 0.007	99.978 ± 0.007	63.8 ± 0.5	99.975 ± 0.003	99.987 ± 0.003
18/5 mW cm^−2^	ND	ND	ND	66.2 ± 0.5	99.988 ± 0.007	99.99 ± 0.01
18/10 mW cm^−2^	55.8 ± 0.1	99.95 ± 0.06	99.955 ± 0.005	66.5 ± 0.4	99.75 ± 0.06	99.91 ± 0.01
27/10 mW cm^−2^	57.8 ± 0.1	99.999 ± 0.003	lethal	68 ± 2	99.985 ± 0.004	99.986 ± 0.004
36/10 mW cm^−2^	57.38 ± 0.03	ND	ND	70 ± 3	99.986 ± 0.004	99.99 ± 0.02

ND—not determined.

## Data Availability

Data is contained within the article.

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
