# Peer review of "Biogenic Gold Nanoparticles Decrease Methylene Blue Photobleaching and Enhance Antimicrobial Photodynamic Therapy"

_molecules, 2021, doi:10.3390/molecules26030623_

Round 1

Reviewer 1 Report

This paper describes the development of a dual AuNP/photosensitised strategy for both G+ and G- antibacterial activity.  This represents a significant improvement on earlier work which was more limited to G+ activity; the effectiveness would appear to derive from the homogeneity and purity of the bioderived AuNPs.  The study of antimicrobial efficacy is very thorough and there are only a few presentational matters which require modification.

Comments:

Table 1 is MB but without AuNPs - please clarify

Figure 2, 3, 4 - what is the control, please indicate in table sub-text

Lines 251-255 repeat lines 240-245; english construction here is very clumsy

Author Response

thank you very much for your letter concerning our article entitled „ Biogenic gold nanoparticles decrease Methylene Blue photobleaching and enhance antimicrobial photodynamic therapy”

We would like to thank the Referees for comments. We answered each point and the detailed answers are enclosed. We hope that our article will be reconsidered for publication in “Molecules ” journal.  

Reviewer#1.

(1) Table 1 is MB but without AuNPs - please clarify

The results of dark toxicity of  AuNPs and MB were omitted, as the observed inhibition of bacterial growth was insignificant. These data were added to the manuscript (yellow mark on page 4), as it has been suggested.

(2) Figure 2, 3, 4 - what is the control, please indicate in table sub-text

Control is the initial concentration of bacteria in the suspensions kept in the dark; this note has been indicated under tables 2, 3 and 4, as it has been suggested.  

(3) Lines 251-255 repeat lines 240-245; english construction here is very clumsy

The repeated lines have been deleted. A new description was included (yellow mark).

Reviewer 2 Report

In this paper, the authors studied the effect of biogenic gold nanoparticles to prevent the methylene blue photobleaching, enhancing its  photoactivity and bactericidal effect. The work is clear and well-structured and the achieved results are important for the development of photodynamic therapy. Therefore, I can recommend this manuscript for publication after the following revision:

-In page 3,  lines 106-109, the authors wrote “…cells of S. aureus were 61±3%, 45±4%, 28±4%, 18±3%, 11±2% and 8±2% for MB concentration of 250 106 mgL-1, 100 mgL-1, 50.0 mgL-1, 25 mgL-1, 12.5 mgL-1 and 6.25 mgL-1, respectively. In the case of E. coli 107 the reduction in viability was 45±4%, 15±3%, 9±2%, 7±2%, 5±1% , 3±1% for MB concentration of 250 108 mgL-1, 100 mgL-1, 50.0 mgL-1, 25 mgL-1 12.5 mgL-1 and 6.25 mgL-1.”, please do not repeated the unities and just put only the units in the last value. Please do similar corrections throughout the manuscript ( see pages 4, 5, 6, …)

-The kinetic curves of MB photobleaching in absence of gold nanoparticles does not follow a linear curve. Please correct and interpret.

- Please add error bars to values of table 2.

- In page 11, lines 11-18, the authors wrote “The mechanism responsible for enhancement of the photo-bactericidal effect by gold nanoparticles has been discussed in the literature.”, references should be indicated.

- In page 11, the authors claim “The leuco form of MB, produced by adding a hydride anion, is most often mentioned as a candidate for a transparent product (Figure 8) [39, 40]. It is believed that the two-step character [36] of the bleaching process observed under aerobic conditions is consistent with the formation of the leuco-MB form as the main product [40]. This meta-stable structure is quenched by molecular oxygen, and it has been considered that part of this quenching process leads to a subsequent chemical modification of the chromophore [41]. Clarify this set of sentences and specially the last sentence. Which is the chemical modification? A scheme about the effect of radiation on presence of gold nanoparticles and MB molecules and respective modifications on the molecules will be would be very welcome to the readers.

Author Response

(1) In page 3,  lines 106-109, the authors wrote “…cells of S. aureus were 61±3%, 45±4%, 28±4%, 18±3%, 11±2% and 8±2% for MB concentration of 250 106 mgL-1, 100 mgL-1, 50.0 mgL-1, 25 mgL-1, 12.5 mgL-1 and 6.25 mgL-1, respectively. In the case of E. coli 107 the reduction in viability was 45±4%, 15±3%, 9±2%, 7±2%, 5±1% , 3±1% for MB concentration of 250 108 mgL-1, 100 mgL-1, 50.0 mgL-1, 25 mgL-1 12.5 mgL-1 and 6.25 mgL-1.”, please do not repeated the unities and just put only the units in the last value. Please do similar corrections throughout the manuscript ( see pages 4, 5, 6, …)

This adjustment was introduced throughout the manuscript, as it has been suggested.   

(2) The kinetic curves of MB photobleaching in absence of gold nanoparticles does not follow a linear curve. Please correct and interpret.

The kinetics of the photobleaching of MB and MB+AuNPs seems to be much more complicated than we initially assumed (as a 1st order kinetics) thus we removed the information about the kinetics studies and show only the change in the absorbance spectra. The studies of the kinetics will be further developed as a separate work.

 (3) Please add error bars to values of table 2.

Error bars to values have been added in Table 2, as it has been suggested.

(4) In page 11, lines 11-18, the authors wrote “The mechanism responsible for enhancement of the photo-bactericidal effect by gold nanoparticles has been discussed in the literature.”, references should be indicated.

The references were added (positions 35-37), as it has been suggested.

(5) In page 11, the authors claim “

This meta-stable structure is quenched by molecular oxygen, and it has been considered that part of this quenching process leads to a subsequent chemical modification of the chromophore [41].

Clarify this set of sentences and specially the last sentence. Which is the chemical modification? A scheme about the effect of radiation on presence of gold nanoparticles and MB molecules and respective modifications on the molecules will be would be very welcome to the readers.

Round 2

Reviewer 2 Report

The authors have improved the manuscript in accordance with the referees comments, so, I can recommend this manuscript for publication.